# Spectrally Selective Full-Color Single-Component Organic Photodetectors Based on Donor-Acceptor Conjugated Molecules

**DOI:** 10.3390/molecules28010368

**Published:** 2023-01-02

**Authors:** Artur L. Mannanov, Dmitry O. Balakirev, Elizaveta D. Papkovskaya, Alexander N. Solodukhin, Yuriy N. Luponosov, Dmitry Yu. Paraschuk, Sergey A. Ponomarenko

**Affiliations:** 1Enikolopov Institute of Synthetic Polymeric Materials, Russian Academy of Sciences, Profsoyuznaya St. 70, 117393 Moscow, Russia; 2Faculty of Physics, Lomonosov Moscow State University, Leninskie Gory 1, 119991 Moscow, Russia

**Keywords:** donor-acceptor molecules, conjugated molecules, organic synthesis, single-component organic photodetectors, human eye

## Abstract

Photodetectors based on organic materials are attractive due to their tunable spectral response and biocompatibility, meaning that they are a promising platform for an artificial human eye. To mimic the photoelectric response of the human eye, narrowband spectrally-selective organic photodetectors are in great demand, and single-component organic photodetectors based on donor-acceptor conjugated molecules are a noteworthy candidate. In this work, we present single-component selective full-color organic photodetectors based on donor-acceptor conjugated molecules synthetized to mimic the spectral response of the cones and rods of a human eye. The photodetectors demonstrated a high responsivity (up to 70 mA/W) with a response time of less than 1 µs, which is three orders of magnitude faster than that of human eye photoreceptors. Our results demonstrate the possibility of the creation of an artificial eye or photoactive eye “prostheses”.

## 1. Introduction

Organic photodetectors have many advantages over conventional inorganic ones, as follows: they can be made from a wide choice of materials, they can be light, thin, flexible, or semitransparent, are easy to manufacture and compatible with printing technologies, and they are biocompatible [1,2,3,4]. The latter is especially important for the possibility of prosthetics of human visual organs [5,6]. An organic photodetector and a solar cell are, per se, the same device used in different operating modes. The most efficient organic solar cells are based on a phase-separated blend of donor and acceptor materials, the optical absorption spectra of which differ and complement each other for the greatest overlap with the solar spectrum [7,8,9]. This circumstance probably explains the greater number of publications devoted to broadband organic photodetectors compared to narrowband ones [10,11]. The disadvantage of broadband photodetectors is the difficulty of color discrimination. As a result, one has to use appropriate color filters to eliminate unwanted sensitivity. The use of color filters complicates the design of the device, and also reduces the photodetector responsivity because of absorption of a part of the incident light [12,13]. Therefore, the development of narrowband photodetectors is of increased interest.

For the implementation of a filter-free narrowband photodetector, various approaches are distinguished, as follows: the development of new narrowband absorbers (both donor and acceptor), the use of nearly transparent donor or acceptor materials, and modification of the architecture of the device to optimize its spectral response [14]. Following the first path, a single donor-acceptor material with a narrow absorption spectrum and effective charge photogeneration seems to be feasible. Recently, much attention has been paid to single-component solar cells based on donor-acceptor materials [15,16]. At the same time, single-component organic photodetectors have far less studied, as follows from articles [17,18,19,20,21,22,23,24]. Single-component organic photodetectors benefit primarily from their simplicity, which is especially important for biological applications. It is also worth noting that single-component photodetectors are more stable compared to photodetectors based on donor-acceptor blends, since the morphology of the latter may change during the device operation due to thermodynamic instability [25].

Conjugated polymers and small molecules are used as materials for single-component photodetectors. The latter combine the advantages of a well-defined chemical structure, ease of purification, and reproducibility, which provides almost identical characteristics of the devices from batch to batch, as compared with the polymers [26]. The donor-acceptor concept in the molecular design of the π-conjugated system opens wide possibilities for tuning their key properties, such as absorption spectra, charge-carrier mobilities, charge photogeneration efficiency, and stability [26,27]. Changing the donor or/and acceptor group, as well as the type and the length of the conjugated π-spacer between them, allows for adjusting the frontier orbitals energies, absorption and luminescence spectra, phase behavior, photophysics, and charge separation [23,24,28]. Various donor-acceptor conjugated molecules of a linear and star-shaped structure were synthesized, and their physicochemical and photovoltaic properties were investigated [23,24,28,29,30]. If the molecular absorption spectra are close to the photosensitive response of eye photoreceptors (blue cones, rods, green cones, red cones), they can be used to create full-color retinal prostheses [6].

In this work, oriented on a natural optical device—the human eye—we found several small π-conjugated donor-acceptor molecules with different optical properties, aiming to reproduce as accurately as possible the absorption spectra of rods and cones of the human eye retina (Figure 1). Prototypes of single-component organic photodetectors based on the most promising materials were fabricated and characterized. Full-color spectrally-selective single-component organic photodetectors that completely mimic human vision are presented for the first time.

## 2. Results and Discussion

### 2.1. Synthesis of the Materials

Figure 2 shows the chemical structures of the molecules investigated. These materials were chosen because the maxima of their absorption spectra in thin films correspond well to the maxima of the sensitivity of the human eye photoreceptors (Figure 1). Synthesis and investigation of the properties of molecules **II** and **IV** were described previously [32,33,34]. Synthesis and characterization of **I** and **III** are described below (Figure 1). 

The novel TPA-based star-shaped molecules **I** and **III** were prepared similar to the previously developed synthetic approach based on preparation of the star-shaped triketones or trialdehydes followed by a Knövenagel condensation [35].

Synthesis of **I** was carried out via Knövenagel condensation reaction of compound **1** and excess of 2-(1-oxopropoxy)acetonitrile in pyridine (Figure 1a) to give **I** in 72 % yield. The detailed synthetic procedures and characterization data can be found in the Appendix A.

Synthesis of **III** consisted of three consecutive reaction stages. First, the preparation of ketone precursor **3** by the acylation reaction of 4-bromo-7-(2-thienyl)-2,1,3-benzothiadiazole **2** and heptanoyl chloride using tin tetrachloride as a catalyst in 82 % yield was carried out. The second stage included the synthesis of the star-shaped compound **5**, which was prepared through a Suzuki cross-coupling reaction between 4,4′,4″-*tris*(4,4,5,5-tetramethyl-1,3,2-dioxaborolan-2-yl)triphenylamine **4** and ketone precursor **3** in 65 % yield. Finally, **III** was obtained by Knövenagel condensation of compound **5** with the excess of malononitrile in pyridine using a microwave heating to give the product in 53% yield. 

Here, ^1^H- and ^13^C-NMR spectroscopy, elemental analysis, and MALDI-TOF were used to characterize the molecular structure and purity of these molecules (see Appendix A). All target molecules were readily soluble in common organic solvents, such as THF, chloroform, dichloromethane, etc.

### 2.2. Optical Properties of the Materials

Figure 3 shows the normalized absorption spectra of compounds **I**–**IV** in thin films (normalized absorption spectra in THF solution are presented in the Appendix A). The spectra of all compounds in thin films show good overlap with those of eye photoreceptors, as the differences in spectral maxima positions do not exceed 10 nm (25 nm for blue cones). At the same time, it is worth noting that the positions of the sensitivity maxima of rods and cones of different people vary within ±10 nm [36].

### 2.3. Charge Transport

Since the photoactive layer of spectrally-selective photodetectors usually consists of just one donor-acceptor material, its hole and electron mobilities should be large enough and comparable to ensure fast, effective, and balanced charge transport of the photogenerated charges to the device electrodes. The high and close values of the hole and electron mobilities should be beneficial for photodetectors to avoid space charge effects and increase the operation speed. The hole and electron mobilities in films of molecules **I**–**IV** was measured by the space charge limited current (SCLC) technique (for details see the Materials and Methods section), and are presented in Table 1 (*J*-*V* characteristics of the hole and electron only devices are presented in Appendix A). Materials **I**–**IV** demonstrated a sufficiently balanced charge transport, and the charge mobility values are able to provide a response time for photodetectors with the photoactive layer thickness of ~50 nm in the sub-microsecond range.

### 2.4. Responsivity Spectra

Single-component photodetectors based on the donor-acceptor materials studied were fabricated (experimental details are described in the Materials and Methods section), and their main characteristics [1,2,11] were measured (the measurement details are described in the Materials and Methods section), such as the responsivity (*R*) spectra, response speed, dark current, and specific detectivity. The corresponding *J*-*V* curves, photovoltaic parameters, and external quantum efficiency (EQE) spectra were also measured and presented in the Appendix A.

Since the intrinsic charge photogeneration in material **IV** is not very efficient, a small amount of an acceptor (PC_71_BM, no more than 10%) was added to it. In this case, there is a significant improvement in the device characteristics (maximum *R* and EQE) with virtually no change in the shape of the *R* and EQE spectra. The measured and normalized *R* spectra are presented in Figure 4, and the spectral sensitivity maxima are summarized in Table 2.

All responsivity spectra of the photodetectors (with the exception of the blue ones) show a slight redshift for the peak wavelengths compared to the absorption spectra of the same films (Figure 3), which is most likely due to the effects of optical resonator in the device. Nevertheless, the responsivity spectra correspond well to the spectral response of human cones and rods (Figure 1). All the photodetectors demonstrated a difference in peak wavelength within 25 nm as compared to the photoresponse spectra of cones and rods. At the same time, it is worth noting that positions of the sensitivity maxima of rods and cones of different people vary within ±10 nm [36]. This means that an exact match between the photodetectors photoresponse and the absorption of human cones and rods may not be necessary. The green and red photodetectors showed the highest responsivity values, while the blue photodetectors showed a lower responsivity as compared to the others. Such characteristics of photodetectors fabricated somewhat resemble the behavior of human photoreceptors, where the highest response is observed for green cones, and for blue cones it is lower.

### 2.5. Time-Resolved Response

Figure 5 shows the time-resolved response of the photodetectors to a rectangular optical pulse (the measurement details are described in the Materials and Methods section). The rise-times of the photodetector responses are summarized in Table 2, as follows: they are shorter than 1 µs, which is three orders of magnitude faster than the response time of human eye photoreceptors. The main factors that have a decisive influence on the rise-time are the following: the charge mobility values, exciton dynamics, charge generation rate, applied voltage, and RC time constant [37]. The exciton lifetime in a film of **II** is significantly lower than 1 ns [24]. Therefore, the delayed charge generation at a sub-microsecond time scale is unlikely to have a significant impact on the response time. The RC time constant is also in the nanosecond time range (see the Materials and Methods section) and should have no effect on the response time. We believe that the main factor determining the response time is the charge carrier mobility. Indeed, the time (*τ*) of flight of a charge carrier through the thickness (*d*) of the active layer is estimated as *τ* = *d*/(*µF)*, where *F* is the internal electric field. As a result, the time of flight is in the sub-microsecond range (0.15–0.3 µs).

### 2.6. Photodetectors Specific Detectivity

Table 3 presents the results on the dark current density (*J*_dark_) and specific detectivity (*D**) of reverse biased photodetectors. The specific detectivity values in the generally accepted assumption of the dominance of shot noise under reverse bias conditions [38,39] were calculated using the following Equation (1):(1)$$D^{*}=\frac{R}{\sqrt{2qJ_{\mathrm{dark}}}}$$

Relatively large dark currents of 50–1200 nA/cm^2^ are a consequence of a small thickness of the photoactive layer. As a result, the specific detectivity values were reduced because of the large dark currents. The thickness of the single-component photodetectors (see the Materials and Methods section) is optimal in terms of the highest *R* and EQE values. The thickness of the photoactive layer is limited by the relatively low values of the charge carrier mobility (Table 1) of the organic semiconductor materials used. Therefore, a popular approach is to reduce the dark current by increasing the photoactive layer thickness to achieve a higher shunt resistance, although this faces a trade-off between high values of EQE, *R*, response speed, *D**, and low *J*_dark_ values. Although the specific detectivity values turned out to be small, they are not inferior, and sometimes even superior to those for other recently published single-component photodetectors [17,18,19,20].

## 3. Materials and Methods

### 3.1. Photodetectors Fabrication and Characterization

The structure of the photodetectors with normal architecture was ITO/PEDOT:PSS/photoactive layer/Ca/Al. The samples of photodetectors were prepared in the following way. First, glass substrates coated with a patterned indium-tin oxide (ITO) layer (Xin Yan Technology Limited, Kwun Tong, Kowloon, China) were cleaned in an ultrasonic bath with a surfactant and then exposed to an ultraviolet lamp (Photo Surface Processor PL16-110, 15 mW/cm^2^ isopropanol (Bandelin sonorex digitec, BANDELIN electronic GmbH & Co. KG, Berlin, Germany), and were, 254 nm) (Sen Lights Corporation, Osaka, Japan). Then a 50 nm-thick layer of poly(ethylenedioxythiophene):polystyrene sulfonate (PEDOT:PSS, CLEVIOS P VP AI 4083, Heraeus GmbH, Leverkusen, Germany) was deposited on the ITO by spin-coating at 3000 rpm and annealing at 140 °C for 15 min. The photoactive layer was deposited on the PEDOT:PSS layer by doctor blading from chloroform solution with a concentration of 10 g/L (substrate temperature 50 °C, blade speed 15 mm/s); the resulting thickness was 50 ± 5 nm. On top of the photoactive layer, a Ca (20 nm)/Al (80 nm) electrode was evaporated under pressure less than 5·10^−6^ mbar with the use of a vacuum evaporator (Univex 300, Leybold, Köln, Germany) integrated in a glove box with an Ar atmosphere (H_2_O < 0.1 ppm, O_2_ < 5 ppm). Eight devices with the active area of each device being 3 mm^2^ were formed on substrates by using a shadow mask.

Inverted architecture of photodetectors was chosen for response time measurements due to the higher current. The structure of the photodetectors with inverted architecture was ITO/ZnO/photoactive layer/MoO_3_/Ag. The samples of inverted photodetectors were prepared similar to normal photodetectors. The ZnO layer was deposited on the ITO layer in the following way. A total of 100 mg of zinc acetate dihydrate (Merck KGaA, Darmstadt, Germany) was dissolved in a mixture of 1 mL of 2-methoxyethanol (Merck KGaA, Darmstadt, Germany) and 27 µL of monoethanolamine (Merck KGaA, Darmstadt, Germany). Then, this solution was bladed on the ITO layer and annealed at 140 °C for 25 min, resulting in a 40 nm ZnO layer. The MoO_3_ (10 nm)/Ag (100 nm) electrode was deposited on the photoactive layer with the use of a vacuum evaporator (Univex 300, Leybold, Köln, Germany). Eight devices with inverted architecture and a 3 mm^2^ active area for each device were formed on substrates by using a shadow mask.

The *J*-*V* characteristics of photodetectors were measured using a source-meter (SourceMeter 2400, Keithley, Tektronix, Inc., Beaverton, OR, USA). Responsivity (*R*) and external quantum efficiency (EQE) spectra were measured with the use of a laser-driven light source (LDLS EQ-99X, Energetiq Technology, Inc., Woburn, MA, USA) equipped with a monochromator (CS130-USB-3-MC, Newport, Irvine, CA, USA), a power sensor (S120UV, Thorlabs, New Jersey, NJ, USA), and a source-meter (SourceMeter 2400, Keithley, Tektronix, Inc., Beaverton, OR, USA). To avoid higher diffraction orders, additional filters KG3, GG400, and OG550 (Newport, Irvine, CA, USA) were used for the 380–500, 480–620 and 600–800 nm spectral ranges, respectively. All the measurements were performed in a glove box with Ar atmosphere (H_2_O < 0.1 ppm, O_2_ < 5 ppm).

### 3.2. ChargeCarrier Mobility Measurements

Charge carrier mobility was determined using space charge limited current (SCLC) measurements on unipolar thin-film devices of several thicknesses. The structure of the hole-only devices was ITO/PEDOT:PSS/active layer/MoO_3_/Ag. The structure of the electron-only devices was ITO/ZnO/active layer/Ca. Unipolar devices were made similarly to photodetectors. The hole and electron mobilities were extracted by fitting the current-voltage (*J*-*V*) characteristics of the corresponding devices with the simplest SCLC model. According to the SCLC model, and taking into account the series (*R*_s_) and shunt (*R*_sh_) resistances (as fitting parameters), the charge mobility was calculated by approximating the dark *J*-*V* curves of unipolar devices using the following Equation (2):(2)$$J=\frac{9}{8}\varepsilon \varepsilon_{0}\mu \frac{{\left(V-V_{\mathrm{BI}}-JSR_{s}\right)}^{2}}{d^{3}}+\frac{V-JSR_{s}}{R_{\mathrm{sh}}}$$
where *ε*_0_ = 8.85·10^−12^ F/m, *ε* is the dielectric permittivity (taken as 3), *d* is the active layer thickness (measured by AFM), and *V*_BI_ is the built-in voltage (fitting parameter).

### 3.3. Time-Resolved Response of Photodetectors

Rise-time measurements were performed using a green laser diode (505 nm, for photodetectors based on **I**–**III**) (Sharp GH05130C2GL, Osaka, Japan) and blue laser diode (450 nm, for photodetectors based on **IV**) (Sharp GH04C01A2G, Osaka, Japan) controlled by a laser driver (LDP-V 03-100 V4.0, PicoLAS, Würselen, Germany) equipped with a pulse generator (PLCS-21, PicoLAS, Würselen, Germany). Rectangular optical pulses (rise-time less than 1.2 ns) with a pulse length of 10 µs and repetition rate of 1 kHz illuminated the active area of the photodetector (without voltage bias) with an average incident power of 110 µW (green laser) and 700 µW (blue laser). The photodetector was connected to a Tektronix TDS 1012B oscilloscope with a 50 Ω resistor connected in parallel. The transients were obtained via an OpenChoice program by Tektronix. The rise-time was taken as the time during which the electrical response of the device rose from 10% to 90%. The corresponding RC-time constant of the electronics was 2–20 ns, and did not impact on the measured rise-time.

## 4. Conclusions

Donor-acceptor conjugated molecules with the absorption spectra similar to that of the human eye photoreceptors for single-component organic photodetectors were synthesized. Organic single-component selective full-color photodetectors based on synthesized molecules, mimicking the human eye, were reported for the first time. These photodetectors demonstrated a sufficiently high responsivity at the level of 3–70 mA/W, comparable to the responsivity of the human eye receptors. The spectral selectivity of photodetectors is comparable to that of the corresponding cones and rods of the human eye. The photodetector response time (<1 µs) is one of the fastest among organic photodetectors, as it is three orders of magnitude faster than the response time of a human eye. The results obtained pave the way to creating an artificial eye—a matrix of organic photodetectors with different spectral sensitivities, or photoactive “prostheses” for an eye devoid of sensitivity to light because of diseases. We expect that our results will stimulate the development of new conjugated donor-acceptor molecules and facilitate further progress in organic single-component selective photodetectors.

## Data Availability

The data presented in this study are available in Appendix A.

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
