# Peer review of "Spectrally Selective Full-Color Single-Component Organic Photodetectors Based on Donor-Acceptor Conjugated Molecules"

_molecules, 2023, doi:10.3390/molecules28010368_

Round 1
Reviewer 1 Report
In this manuscript, "Spectrally selective full-color single-component organic photo-detectors based on donor-acceptor conjugated molecules", Paraschuk, Ponomarenko, and coauthors report the synthesis and characterization of single narrowband spectrally selective organic photodetectors to mimic the human eye photoreceptors. The work looks well-described and clear and the conclusions are supported by the experimental data. I recommend the publication on Molecules after the following minor revisions.
- Captions in figures 3-5 and tables 1-3 are misleading, in my opinion. In the figures, there are no references to the selected molecules; in the tables, the molecules are identified with acronyms, while, in figure 2, the authors use roman numbers. I suggest introducing an unambiguous nomenclature, using the same roman numbers or the same acronyms in all the parts of both the main text and the SI, to help the comprehension by readers.
- In the NMR characterizations: 1) substitute "overlapping peaks" with "multiplet". 2) When authors write "m=5" I guess they intend "quintuplet", so substitute it with "q". 3) In the 13C-NMR of compound 5, a carbon signal (24.86 ppm) which is present in the spectrum is missing in the peaks list. 4) In the "characterization" section, the authors report DMSO-d6, but they do not report any spectrum in that solvent. 5) All the 1H and 13C-NMR spectra should not be cut immediately after the last peak: you should show them at least until 11 ppm for 1H and 230 ppm for 13C-NMR spectra.
Author Response
Reviewer 1
In this manuscript, "Spectrally selective full-color single-component organic photo-detectors based on donor-acceptor conjugated molecules", Paraschuk, Ponomarenko, and coauthors report the synthesis and characterization of single narrowband spectrally selective organic photodetectors to mimic the human eye photoreceptors. The work looks well-described and clear and the conclusions are supported by the experimental data. I recommend the publication on Molecules after the following minor revisions.
We thank the Reviewer for positive evaluation of our manuscript and constructive critical notes that allowed us to improve it.
- Captions in figures 3-5 and tables 1-3 are misleading, in my opinion. In the figures, there are no references to the selected molecules; in the tables, the molecules are identified with acronyms, while, in figure 2, the authors use roman numbers. I suggest introducing an unambiguous nomenclature, using the same roman numbers or the same acronyms in all the parts of both the main text and the SI, to help the comprehension by readers.
In the revised manuscript we use the same nomenclature in roman numbers everywhere: in Figures, Tables and in the text.
- In the NMR characterizations: 1) substitute "overlapping peaks" with "multiplet". 2) When authors write "m=5" I guess they intend "quintuplet", so substitute it with "q". 3) In the 13C-NMR of compound 5, a carbon signal (24.86 ppm) which is present in the spectrum is missing in the peaks list. 4) In the "characterization" section, the authors report DMSO-d6, but they do not report any spectrum in that solvent. 5) All the 1H and 13C-NMR spectra should not be cut immediately after the last peak: you should show them at least until 11 ppm for 1H and 230 ppm for 13C-NMR spectra.
Designations for the NMR characterization were corrected taking into account the reviewer’s comments: "overlapping peaks" was changed with "m", "m=5" was changed with "q". Misprints with DMSO-d6 report and missing signal as well as the 1H and 13C-NMR spectra intervals have also been corrected.
All changes in the manuscript are marked up using the “Track Changes” function of MS Word. Beside these changes, some minor changes (spelling, grammar, formatting etc.) were done in the revised manuscript. All references are given for the revised manuscript.
We hope that re revised manuscript can be accepted for a publication.
Reviewer 2 Report
Organic photodetectors are attractive due to their tunable spectral response and biocompatibility. In this contribution, the author demonstrated a high responsivity with the response time less than 1 µs, which is three orders of magnitude faster than that of human eye photoreceptors. This manuscript has shown a complete systematic story with reasonable conclusions.
Most reported organic photodetectors are based on electron donor and acceptor blends. What is the advantage of single-component organic photodetectors?
Author Response
Reviewer 2
Organic photodetectors are attractive due to their tunable spectral response and biocompatibility. In this contribution, the author demonstrated a high responsivity with the response time less than 1 µs, which is three orders of magnitude faster than that of human eye photoreceptors. This manuscript has shown a complete systematic story with reasonable conclusions.
We thank the Reviewer for positive evaluation of our manuscript.
Most reported organic photodetectors are based on electron donor and acceptor blends. What is the advantage of single-component organic photodetectors?
In the initial manuscript, at the end of the first paragraph of the Introduction, we mentioned the disadvantages of broadband organic photodetectors, which are most often based on donor-acceptor blends. In addition to the fact that narrowband single-component organic photodetectors are devoid of these disadvantages, they benefit primarily from their simplicity, which is especially important for biological applications. It is also worth to note that donor-acceptor blends may be unstable, since their morphology may change during the device operation due to thermodynamic instability [25], therefore single-component photodetectors are more stable. In the revised manuscript we have added this point at the end of the second paragraph of the Introduction.
All changes in the manuscript are marked up using the “Track Changes” function of MS Word. Beside these changes, some minor changes (spelling, grammar, formatting etc.) were done in the revised manuscript. All references are given for the revised manuscript.
We hope that re revised manuscript can be accepted for a publication.